# Identifying high risk clinical phenogroups of pulmonary hypertension through a clustering analysis

Paula Rambarat[1⚬], Emily K. Zern[2⚬], Dongyu Wang[3], Athar Roshandelpoor[3], Shahrooz Zarbafian[2], Elizabeth E. Liu[3], Jessica K. Wang[3], Jenna N. McNeill[4], Carl T. Andrews[2], Eugene V. Pomerantsev[2], Nathaniel Diamant[5], Puneet Batra[5], Steven A. Lubitz[6], Michael H. Picard[2], Jennifer E. Ho[3]*

1 Department of Medicine, Massachusetts General Hospital, Boston, Massachusetts, United States of America, 2 Division of Cardiology, Massachusetts General Hospital, Boston, Massachusetts, United States of America, 3 Cardiovascular Institute and Division of Cardiology, Department of Medicine, Beth Israel Deaconess Medical Center, Boston, Massachusetts, United States of America, 4 Division of Pulmonary and Critical Care, Massachusetts General Hospital, Boston, Massachusetts, United States of America, 5 Broad Institute of the Massachusetts Institute of Technology and Harvard, Cambridge, Massachusetts, United States of America, 6 Cardiovascular Research Center, Massachusetts General Hospital, Boston, Massachusetts, United States of America

⚬ These authors contributed equally to this work.
* jho@bidmc.harvard.edu

## Abstract

### Introduction

The classification and management of pulmonary hypertension (PH) is challenging due to clinical heterogeneity of patients. We sought to identify distinct multimorbid phenogroups of patients with PH that are at particularly high-risk for adverse events.

### Methods

A hospital-based cohort of patients referred for right heart catheterization between 2005–2016 with PH were included. Key exclusion criteria were shock, cardiac arrest, cardiac transplant, or valvular surgery. K-prototypes was used to cluster patients into phenogroups based on 12 clinical covariates.

### Results

Among 5208 patients with mean age 64±12 years, 39% women, we identified 5 distinct multimorbid PH phenogroups with similar hemodynamic measures yet differing clinical outcomes: (1) "young men with obesity", (2) "women with hypertension", (3) "men with overweight", (4) "men with cardiometabolic and cardiovascular disease", and (5) "men with structural heart disease and atrial fibrillation." Over a median follow-up of 6.3 years, we observed 2182 deaths and 2002 major cardiovascular events (MACE). In age- and sex-adjusted analyses, phenogroups 4 and 5 had higher risk of MACE (HR 1.68, 95% CI 1.41–2.00 and HR 1.52, 95% CI 1.24–1.87, respectively, compared to the lowest risk phenogroup

**Data Availability Statement:** Data cannot be shared publicly because of potentially identifiable information linked to the electronic health record. Data are available from the Mass General Brigham

Data Access / Ethics Committee
(RPDRHelp@partners.org) for researchers who
meet the criteria for access to confidential data.

**Funding:** JEH was supported by NIH grants R01-
HL134893, R01-HL140224, and K24-HL153669.
The funders had no role in study design, data
collection and analysis, decision to publish, or
preparation of the manuscript.

**Competing interests:** JEH has received past
research grant support from Bayer AG. PB and ND
have received past research support from Bayer
AG and IBM research. SAL has received research
support from Bristol Myers Squibb, Pfizer, Bayer
AG, Boehringer Ingelheim and Fitbit, IBM, and is an
employee of Novartis. This does not alter out
adherence to PLOS ONE policies on sharing data
and materials.

1). Phenogroup 4 had the highest risk of mortality (HR 1.26, 95% CI 1.04–1.52, relative to phenogroup 1).

## Conclusions

Cluster-based analyses identify patients with PH and specific comorbid cardiometabolic and cardiovascular disease burden that are at highest risk for adverse clinical outcomes. Interestingly, cardiopulmonary hemodynamics were similar across phenogroups, highlighting the importance of multimorbidity on clinical trajectory. Further studies are needed to better understand comorbid heterogeneity among patients with PH.

## Introduction

Pulmonary hypertension (PH) is a symptomatic, progressive disorder with high morbidity and mortality [1]. Once diagnosed by an elevated mean pulmonary pressure > 20 mmHg, PH can be classified into one of five World Symposium on Pulmonary Hypertension (WSPH) groups based on etiology, which then informs disease prognosis and management [1]. In practice, however, multimorbidity including concomitant cardiopulmonary disease is common which can make defining the etiology of elevated pulmonary pressures more challenging. In addition, overlap between WSPH groups often exists. For instance, comorbid left-sided valvular disease may lead to elevated pulmonary capillary wedge pressure, which creates diagnostic uncertainty in the presence of pre-capillary PH [2]. While therapies exist for pre-capillary disease, these treatments are not indicated for patients with PH driven by left heart failure [1]. In addition to making the diagnosis and management of PH more complex, comorbid conditions also influence PH disease trajectory. This has been most well studied in patients with pulmonary arterial hypertension (PAH), where common clinical comorbidities such as diabetes and obesity have been shown to affect mortality [3–5]. How clinical heterogeneity affects disease trajectory across the spectrum of PH remains unclear.

In this paper, we set out to better understand comorbid heterogeneity in a broad PH cohort using a clustering analysis approach. We hypothesized that 1) a clustering analysis could identify distinct multimorbid phenogroups of patients with PH, and 2) the identified phenogroups would be at differential risk for adverse outcomes. The identification of high-risk PH phenogroups might help clinicians recognize which patients warrant expedited management and close follow-up. Furthermore, identifying distinct patient groups that share a clinical trajectory beyond current WSPH disease etiology might elucidate mechanisms important to the pathogenesis of PH.

## Methods

### Study sample

We examined consecutive ambulatory and hospitalized patients undergoing a clinically indicated right heart catheterization (RHC) at Massachusetts General Hospital (MGH) between 2005 and 2016 (n = 16788). In cases where patients had multiple RHC procedures, only the initial RHC was included in the analysis (n = 10306). We excluded individuals with acute myocardial infarction occurring on the same day as the RHC, cardiac arrest or shock within 24 hours of RHC, mechanical ventilation, intra-aortic balloon pump, history of heart or lung transplantation, complicated adult congenital heart disease, valvular replacement, and end-

stage renal disease on dialysis (n = 887). Cases were also excluded if missing key clinical covariates (n = 484), patient identifier variables (n = 398), or hemodynamic parameters (n = 252). Lastly, we excluded all patients with a mean pulmonary arterial pressure (mPAP) ≤ 20 mmHg at rest (n = 3077), leaving a final sample size of 5208 patients with PH for analysis (S1 Fig). The requirement for informed consent was waived by the Mass General Brigham Institutional Review Board which approved the study. All data were fully anonymized before it was accessed.

## Data collection

**Clinical, laboratory, echocardiographic and hemodynamic data.** Clinical characteristics, laboratory data, and comorbidities were ascertained from the medical history as part of clinical assessment done at the time of RHC, except for obstructive sleep apnea (OSA) and atrial fibrillation (AF), which were identified based on *International Classification of Diseases Ninth Revision* (*ICD-9*) or *Tenth Revision* (*ICD-10*) codes. Echocardiographic data was abstracted within one year of RHC. The majority of patients (59%) had an echocardiogram within one month of RHC; only 4% of patients had an echocardiogram longer than 6 months but less than one year within RHC. RV dysfunction was defined based on clinical report of RV dysfunction of at least moderate severity. Hemodynamic data were recorded at the time of the RHC. Cardiac index (CI) was derived via thermodilution (TD) method using the Mosteller formula for body surface area. We classified PH into hemodynamic subgroups according to accepted parameters [6]. Pre-capillary PH was defined as pulmonary vascular resistance (PVR)≥3 Wood Units (WU) and pulmonary capillary wedge pressure (PCWP)≤15 mmHg, isolated post-capillary PH as PVR<3 WU and PCWP>15 mmHg and combined pre- and post-capillary PH (Cpc-PH) as PVR≥3 WU and PCWP>15 mmHg. Patients with PVR<3 WU and PCWP≤15 mmHg were designated as having "other PH".

**Clinical outcomes.** All-cause mortality was ascertained using the National Social Security Death Master Index and hospital records. The precise dates of deaths that occurred between 06/10/2017 and the date of abstraction (06/10/2020) are protected nationally for confidentiality reasons. Thus, these death events were imputed as occurring midway through this period (on 12/10/2018, n = 151 out of total of 2182 death events). A major adverse cardiac event (MACE) was defined as a composite of heart failure admissions, cerebrovascular accident, transient ischemic attack, or acute myocardial infarction determined by ICD-9 or ICD-10 code as a primary discharge diagnosis; heart failure admissions were additionally identified via Current Procedural Terminology (CPT) code for heart transplantation or durable ventricular assist device. PH admissions were defined by a corresponding ICD-9 or ICD-10 code as primary discharge diagnosis for an inpatient admission (S1 Table). The follow-up period for each patient was defined as time from RHC to event or date of final encounter in the medical record. Patients were censored based on time of last encounter or death.

## Statistical methods

**Clustering analysis.** K-prototypes [7], an unsupervised partitional clustering algorithm capable of handling both continuous and categorical variables, was used to cluster patients based on 12 baseline clinical covariates including demographics (age, sex, BMI) and clinical comorbidities diagnosed before the date of RHC (heart failure, valvular disease, diabetes, hypertension, prior diagnosis of myocardial infarction (MI), chronic lung disease, OSA, chronic kidney disease not on hemodialysis (CKD), and AF). Covariates were selected *a priori* to represent common clinical variables easily obtained in routine clinical practice. Covariates were dichotomous except for age and BMI which were rescaled to range 0 to 1 before the

cluster analysis. Patients missing key clinical covariates were excluded from the analysis. A silhouette width analysis was used to determine the number of clusters. When examining a range of K = 3 to K = 10 clusters, the average silhouette width was optimal at K = 5 clusters. Clustering was performed using the clustMixType (version 0.2–15) package in R (version 4.1.1).

**Association of phenogroups with clinical outcomes.** We summarized clinical, laboratory, hemodynamic and echocardiographic parameters by phenogroups. Comparison across phenogroups was assessed using analysis of variance (ANOVA), Chi-squared, or Kruskal-Wallis tests as appropriate.

We examined the association between PH phenogroups and all-cause mortality, MACE, and PH hospitalizations using Kaplan-Meier (KM) curves. For time to event analyses, start of follow-up was the date of RHC. Between-group differences were compared formally with log-rank test and Tukey-Kramer correction. We used Cox proportional hazard models adjusted for age and sex to examine the association of phenogroups with outcomes. The proportional hazard assumption was checked using Schoenfeld residuals without major violations. A two-sided p value <0.05 was considered as statistically significant. Statistical analyses were conducted using SAS Version 9.4 (SAS Institute, Cary, NC).

### Cluster stability and internal validation

Clustering stability was assessed using bootstrapping based on the Jaccard coefficient [8, 9] and consensus clustering using the proportion of ambiguous clustering (PAC) [10]. For the former method, bootstrap resampling with 100 randomly selected samples containing 90% of subjects was performed. The mean Jaccard coefficient over all iterations and all clusters was calculated; clustering supported by a Jaccard coefficient $\geq$ 0.75 was treated as robust and stable. For consensus clustering with PAC, we similarly performed 100 clustering runs using random samples containing 90% of subjects. The PAC was calculated from the resulting similarity matrices; values of PAC range from 0 to 1 with a lower value being more stable. Jaccard coefficient and PAC analysis were conducted with the fpc (version 2.2–9) and diceR (version 1.10) packages in R (version 4.1.1).

We also performed internal validation using K = 5 clusters on a randomly selected subset of patients representing 10% of the total study population. Clinical, laboratory, hemodynamic, and echocardiographic data, as well as outcomes, were compared with the original sample.

## Results

### Clinical characteristics of sample

Our study included 5208 patients who underwent a clinically indicated RHC and were found to have mPAP of >20 mmHg at rest. When clustering on clinical comorbidities, we identified five distinct phenogroups. The average age of our sample overall was 64 ± 12 years; 39% were women, and average BMI was 30.2 ± 7.3 kg/m$^2$. There was high prevalence of comorbid cardiopulmonary disease with 61% hypertension, 40% heart failure, 36% valvular disease, 27% diabetes, 22% AF, 21% prior diagnosis of MI, 18% chronic lung disease, and 17% OSA (**Table 1**). Hemodynamic classification of PH showed that 47% of the cohort had post-capillary PH, 14% had pre-capillary PH, 17% had Cpc-PH, and 22% had other PH (PVR<3 and PCWP≤15). Overall, the median mPAP was 29 [24–37] mmHg with a median PVR of 2.2 [1.5–2.3] WU and a median PCWP of 18 [14–24] mmHg (**Table 2**).

**Table 1. Clinical characteristics stratified by phenogroup.**

| | Total (n = 5208) | Young men with obesity (n = 893) | Women with HTN (n = 1060) | Men with overweight (n = 1281) | Men with cardiometabolic/ CVD (n = 1181) | Men with SHD and AF (n = 793) | p value |
|---|---|---|---|---|---|---|---|
| **Phenogroup** | | 1 | 2 | 3 | 4 | 5 | |
| **Demographics** | | | | | | | |
| Age, years | 64 (12) | 45 (9) | 69 (8) | 68 (7) | 65 (9) | 71 (8) | < 0.001 |
| Male sex | 3176 (61) | 576 (65) | 237 (22) | 831 (65) | 889 (75) | 643 (81) | < 0.001 |
| BMI (kg/m$^2$) | 30.2 (7.3) | 30.0 (8.1) | 29.4 (6.9) | 29.6 (7.0) | 33.0 (7.5) | 28.4 (5.8) | < 0.001 |
| **Clinical Comorbidities** | | | | | | | |
| Hypertension | 3150 (61) | 207 (23) | 909 (86) | 250 (20) | 1121 (95) | 663 (84) | < 0.001 |
| Diabetes Mellitus | 1401 (27) | 62 (7) | 213 (20) | 42 (3) | 967 (82) | 117 (15) | < 0.001 |
| Prior MI | 1079 (21) | 62 (7) | 163 (15) | 113 (9) | 491 (42) | 250 (32) | < 0.001 |
| Heart failure | 2057 (40) | 194 (22) | 251 (24) | 113 (9) | 856 (72) | 643 (81) | < 0.001 |
| Valvular disease | 1884 (36) | 168 (19) | 764 (72) | 107 (8) | 226 (19) | 619 (78) | < 0.001 |
| AF | 1128 (22) | 96 (11) | 104 (10) | 223 (17) | 155 (13) | 550 (69) | < 0.001 |
| OSA | 868 (17) | 144 (16) | 117 (11) | 234 (18) | 254 (22) | 119 (15) | < 0.001 |
| Chronic lung disease | 941 (18) | 89 (10) | 220 (21) | 171 (13) | 283 (24) | 178 (22) | < 0.001 |
| CKD | 201 (4) | 14 (2) | 25 (2) | 14 (1) | 108 (9) | 40 (5) | < 0.001 |
| PE | 150 (3) | 38 (4) | 17 (2) | 55 (4) | 26 (2) | 14 (2) | < 0.001 |

Data are expressed as number (%) or average (SD). All demographic and clinical covariates in the table were used for clustering there was no missingness in the data. HTN, hypertension; CVD, cardiovascular disease; SHD, structural heart disease; AF, atrial fibrillation; BMI, body mass index; MI, myocardial infraction; OSA, obstructive sleep apnea; CKD, chronic kidney disease; PE, pulmonary thromboembolism.

## Multimorbidity across phenogroups

We identified five distinct PH phenogroups when clustering on clinical comorbidities (**Table 1**). Phenogroup 1 ("young men with obesity", n = 893) was younger in comparison to other groups (45 ± 9 years) and comprised predominantly of male patients (65%) with obesity (BMI 30.0 ± 8.1 kg/m$^2$). This group had few comorbidities with 23% hypertension. Phenogroup 2 ("women with hypertension", n = 1060) comprised 78% women with age 69 ± 8 years and BMI 29.4 ± 6.9 kg/m$^2$. There was comorbid hypertension and valvular disease in 86% and 72% of patients respectively, but otherwise low prevalence of other cardiopulmonary disease. Phenogroup 3 ("men with overweight", n = 1281) comprised older men (68 ± 7 years) who were overweight (BMI 29.6 ± 7.0 kg/m$^2$) with few comorbidities. Phenogroup 4 ("men with cardiometabolic and cardiovascular disease (CVD)", n = 1181) comprised male patients (75%) with age 65 ± 9 years who were obese (BMI 33.0 ± 7.5 kg/m$^2$). Members of this group had prevalent hypertension (95%), diabetes (82%), heart failure (72%), prior MI (42%), and OSA (22%). Phenogroup 5 ("men with structural heart disease (SHD) and AF", n = 793) comprised older (age 71 ± 8 years), male (81%) patients with high prevalence of hypertension

**Table 2. Laboratory, hemodynamic and echocardiographic characteristics stratified by phenogroup.**

| | Total (n = 5208) | Young men with obesity (n = 893) | Women with HTN (n = 1060) | Men with overweight (n = 1281) | Men with CMD/CVD (n = 1181) | Men with SHD and AF (n = 793) | p value |
|---|---|---|---|---|---|---|---|
| Phenogroup | | 1 | 2 | 3 | 4 | 5 | |
| **Laboratory** | | | | | | | |
| NT-pro BNP (pg/ml) | 2249 [790–5511] | 1694 [586–4683] | 2210 [747–5659] | 2030 [583–5202] | 2113 [758–5119] | 3225 [1492–7429] | < 0.001 |
| **Hemodynamics** | | | | | | | |
| Mean PAP (mmHg) | 29 [24–37] | 30 [24–38] | 28 [24–34] | 29 [24–37] | 30 [25–37] | 31 [26–38] | <0.001 |
| Mean RAP (mmHg) | 9 [6–12] | 9 [6–13] | 8 [5–10] | 8 [6–12] | 10 [7–13] | 9 [7–13] | < 0.001 |
| PCWP (mmHg) | 18 [14–24] | 17 [13–23] | 18 [14–22] | 16 [12–22] | 19 [14–25] | 21 [16–25] | < 0.001 |
| TPG (mmHg) | 11 [8–15] | 11 [8–17] | 10 [8–14] | 13 [9–18] | 11 [8–15] | 10 [7–14] | < 0.001 |
| PVR (WU) | 2.2 [1.5–2.3] | 2.1 [1.4–3.4] | 2.2 [1.6–3.1] | 2.5 [1.6–3.9] | 2.1 [1.4–3] | 2.3 [1.5–3.3] | < 0.001 |
| PAPi | 3.0 [2.0–4.8] | 2.7 [1.7–4.7] | 3.4 [2.3–5.1] | 3.2 [2.1–5.1] | 2.8 [2.0–4.2] | 3.1 [2.1–4.4] | < 0.001 |
| TD CI (L/min/m$^2$) | 2.5 [2.1–3.3] | 2.7 [2.2–3.4] | 2.6 [2.2–3.0] | 2.5 [2.1–3] | 2.5 [2.1–2.9] | 2.3 [1.9–2.7] | < 0.001 |
| **PH Type** | | | | | | | |
| Pre-capillary PH | 647 (14) | 133 (17) | 105 (11) | 256 (22) | 95 (9) | 58 (8) | < 0.001 |
| Post-capillary PH | 2207 (47) | 359 (46) | 463 (48) | 428 (37) | 574 (53) | 383 (55) | < 0.001 |
| Cpc-PH | 809 (17) | 108 (14) | 156 (16) | 196 (17) | 183 (17) | 166 (24) | < 0.001 |
| Other PH | 1037 (22) | 187 (24) | 247 (25) | 278 (24) | 235 (22) | 90 (13) | < 0.001 |
| **Echocardiography** | | | | | | | |
| LVEF (%) | 51 (20) | 49 (22) | 57 (18) | 55 (20) | 45 (20) | 46 (20) | < 0.001 |
| LVEDD (mm) | 51 (10) | 52 (11) | 47 (8) | 48 (9) | 53 (9) | 54 (9) | < 0.001 |
| RV systolic dysfunction (%) | 1085 (21) | 241 (27) | 114 (11) | 267 (21) | 263 (22) | 200 (25) | < 0.001 |
| MR (%) | 1382 (33) | 181 (26) | 338 (39) | 246 (25) | 233 (25) | 384 (56) | < 0.001 |
| TR (%) | 1166 (28) | 158 (22) | 196 (22) | 294 (30) | 212 (24) | 306 (45) | < 0.001 |

Data are expressed as number (%), mean (SD) or median [Q1-Q3]. Pre-capillary PH: PVR≥3WU, PCWP ≤15 mmHg; post-capillary PH: PVR<3WU, PCWP>15 mmHg; combined pre- and post-capillary PH (Cpc-PH): PVR≥3WU, PCWP>15 mmHg; Other PH: PVR<3WU, PCWP≤15 mmHg. Missingness was 26% for both LVEF and LVEDD, and 20% for MR and TR. Right ventricular (RV) systolic dysfunction was defined as diffuse RV hypokinesis on echocardiogram. HTN, hypertension; CMD, cardiometabolic disease; CVD, cardiovascular disease; SHD, structural heart disease; AF, atrial fibrillation; PAP, pulmonary artery pressure; RAP, right atrial pressure; PCWP, pulmonary capillary wedge pressure; TPG, transpulmonary gradient; PVR, pulmonary vascular resistance; PAPi, pulmonary artery pulsatility index; TD, thermodilution; CI, cardiac index; LVEF, left ventricular ejection fraction; LVEDD, left ventricular end-diastolic dimension; RV, right ventricular; MR, mitral regurgitation (moderate or greater); TR, tricuspid regurgitation (moderate or greater).

(84%), heart failure (81%), valvular disease (78%), and AF (69%), (**Fig 1**). When examining presumed WSPH group, based on presence of connective tissue disease (group 1), left heart disease (group 2), lung disease (group 3), and PE (group 4), we found that over 50% of patients

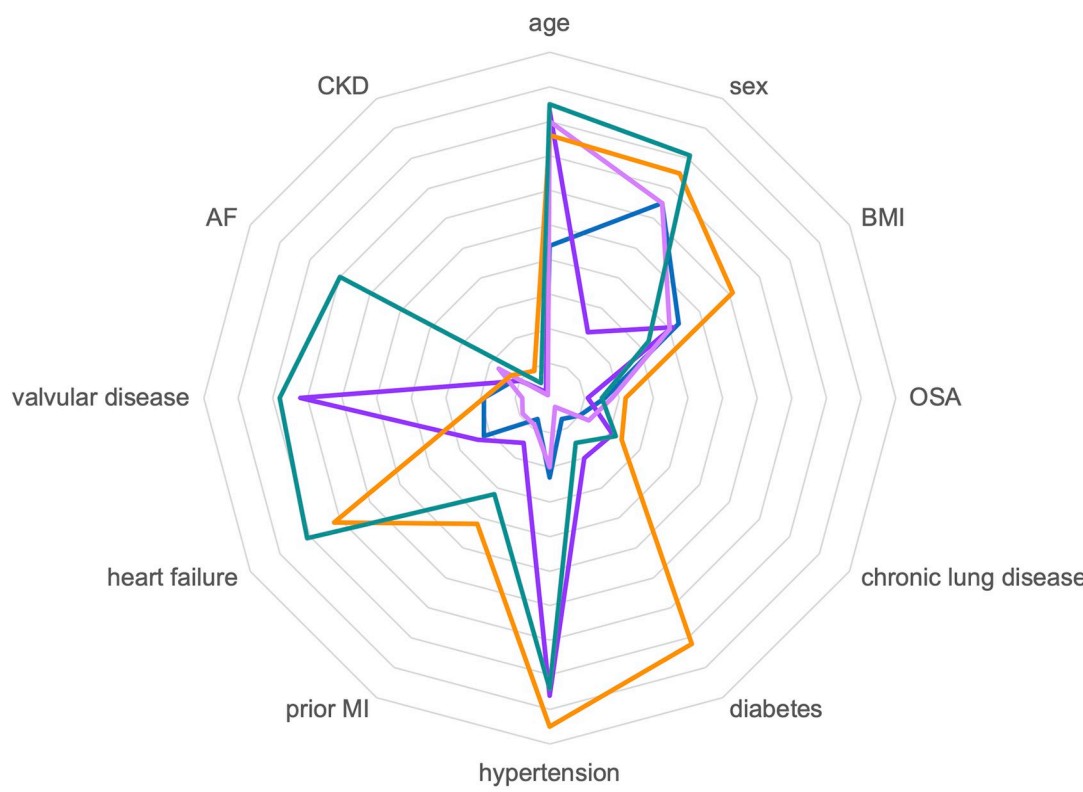

**Fig 1. Differences in demographic and clinical comorbidities across phenogroups.** Age was first rescaled to range 0 to 1 and then the mean plotted on the radar chart. BMI was converted to a dichotomous variable and proportion of patients with BMI ≥ 30 was plotted. Categorical variables are represented as proportions. The axis of the chart ranges from 0 (center) to 1 (in outer perimeter). BMI, body mass index; OSA, obstructive sleep apnea; MI, myocardial infarction; AF, atrial fibrillation; CKD, chronic kidney disease not on hemodialysis.

in clusters 1, 2, and 3 met criteria for 3 or more WSPH groups, whereas the majority of patients in clusters 4 and 5 had left heart disease (S2 Table).

## Hemodynamic measures are similar across phenogroups

There were slight differences in mPAP (median range 28–31 mmHg), right atrial pressure (RAP, median range 8–10 mmHg), PCWP (median range 17–21 mmHg), and transpulmonary gradient (TPG, median range 10–13 mmHg) across phenogroups (Table 2). Despite these minor differences in hemodynamic measures, PH hemodynamic subtypes differed across phenogroups (Fig 2). Phenogroup 3 ("men with overweight") had the highest proportion of pre-capillary PH (22%) and the lowest isolated post-capillary PH (37%), whereas phenogroup 5 ("men with SHD and AF") had the lowest proportion of pre-capillary PH (8%) and the highest isolated post-capillary PH (55%). Right ventricular (RV) function as assessed by the pulmonary

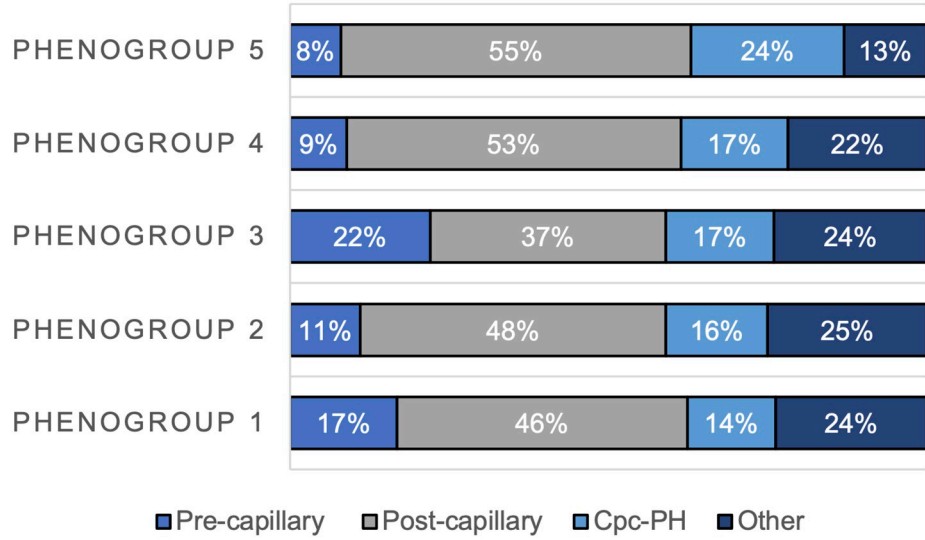

**Fig 2. Analysis of PH hemodynamic type by phenogroup.** Numbers represent percentage of patients in each category. Pre-capillary PH is defined as PVR≥3WU, PCWP ≤15 mmHg; post-capillary PH as PVR<3WU, PCWP>15 mmHg; combined pre- and post-capillary PH (Cpc-PH) as PVR≥3WU, PCWP>15 mmHg, and other PH as PVR<3WU, PCWP≤15 mmHg.

artery pulsatility index (PAPi) [11] was lowest in phenogroup 1 ("young men with obesity") and highest in phenogroup 2 ("women with hypertension"), (**Table 2**).

## Phenogroup differences in cardiac structure and function

While the overall sample had preserved left ventricular (LV) ejection fraction (EF) and normal LV size, individuals in phenogroup 2 ("women with hypertension") had the highest LVEF (57%) and smallest LVEDD (47 mm). By contrast, those in phenogroup 4 ("men with cardiometabolic and CVD") had the lowest LVEF (45%), (**Fig 3**). Phenogroup 1 ("young men with obesity") had the greatest prevalence of RV dysfunction. Lastly, individuals in phenogroup 5 ("men with SHD and AF") had the greatest prevalence of mitral and tricuspid regurgitation, as well as the highest NT-proBNP, whereas those in phenogroup 1 ("young men with obesity") had the lowest NT-proBNP (median levels 3,225 pg/ml versus 1,694 pg/ml, respectively), (**Table 2**).

## Association of phenogroups with adverse outcomes

Over a median follow-up of 6.3 [3.6–9.8] years, there were 2182 deaths, 2002 MACE events, and 376 PH hospitalizations. With respect to all-cause mortality, phenogroup 1 ("young men with obesity") had the highest survival, with the greatest risk of mortality in phenogroups 4 ("men with cardiometabolic and CVD") and 5 ("men with SHD and AF"), (P log rank <0.05, **Fig 4**). For MACE-free survival, phenogroups 1 ("young men with obesity"), 2 ("women with hypertension") and 3 ("men with overweight") appeared similar with lower risk compared to the greatest risk in phenogroup 4 ("men with cardiometabolic and CVD"). Conversely, PH hospitalization-free survival was lowest in phenogroups 1 ("young men with obesity") and 3 ("men with overweight").

In Cox models adjusted for age and sex, phenogroup 1 ("young men with obesity") was chosen as the reference group due to lower absolute incidence of all-cause mortality and

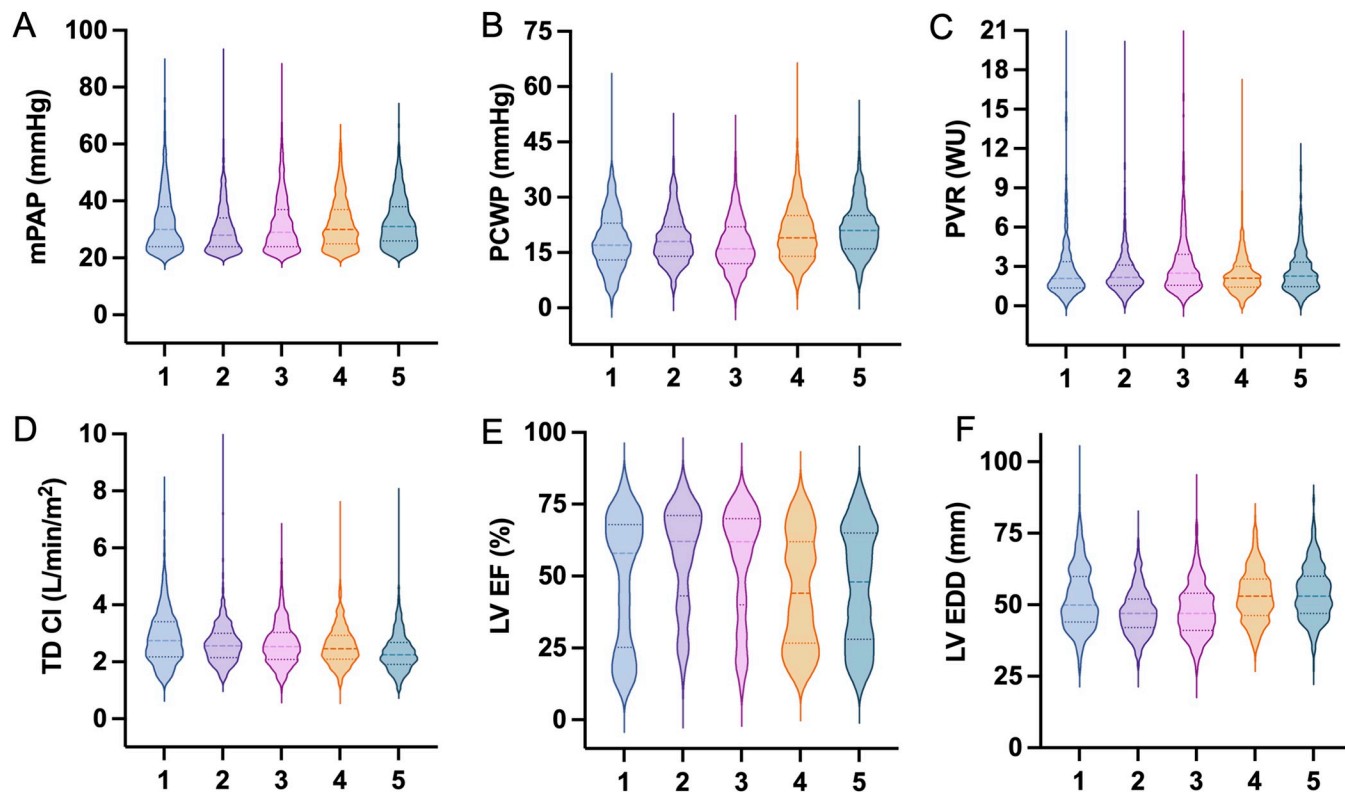

**Fig 3.** Comparison of key hemodynamic (A-D) and echocardiographic (E, F) parameters across phenogroups 1–5 (x-axis). Phenogroup annotations are as follows: Phenogroup 1: "Young Men with Obesity", 2: "Women with Hypertension", 3: "Men with Overweight", 4: "Men with Cardiometabolic and Cardiovascular Disease", 5: "Men with Structural Heart Disease and Atrial Fibrillation". mPAP, mean pulmonary artery pressure; PCWP, pulmonary capillary wedge pressure; PVR, pulmonary vascular resistance; TD, thermodilution; CI, cardiac index; LVEF, left ventricular ejection fraction; LVEDD, left ventricular end diastolic diameter.

MACE. Phenogroup 4 ("men with cardiometabolic and CVD") had 26% higher risk of death compared to phenogroup 1 (HR 1.26, 95% CI 1.04–1.52, p = 0.02). Similarly, phenogroups 4 ("men with cardiometabolic and CVD") and 5 ("men with SHD and AF") had higher risk of MACE compared to phenogroup 1 (HR 1.68, 95% CI 1.41–2.00, p<0.001 and HR 1.52, 95% CI 1.24–1.87, p<0.001, respectively). Conversely, phenogroups 1 ("young men with obesity") and 3 ("men with overweight") had the highest risk of PH hospitalization (**Table 3**).

### Cluster stability and interval validation

The mean Jaccard coefficient was 0.76 indicating acceptable cluster stability. The PAC was 0.49 and was optimal at K = 5 clusters (compared with cluster numbers K = 2–5). For the internal validation of our study, clinical, laboratory, hemodynamic, and echocardiographic characteristics of the 5 phenogroups in the smaller subsample mirrored that of the main analysis, with similar associations with outcomes including lowest risk of MACE in phenogroup 1 and highest risk in phenogroups 4 and 5 (**S3 and S4** Tables).

### Discussion

We performed a clustering analysis on a hospital-based heterogeneous cohort of patients with PH (mPAP >20 mmHg) and found the following: 1) the study cohort had high multimorbidity, low PVR, and high prevalence of isolated post-capillary PH, 2) the five identified

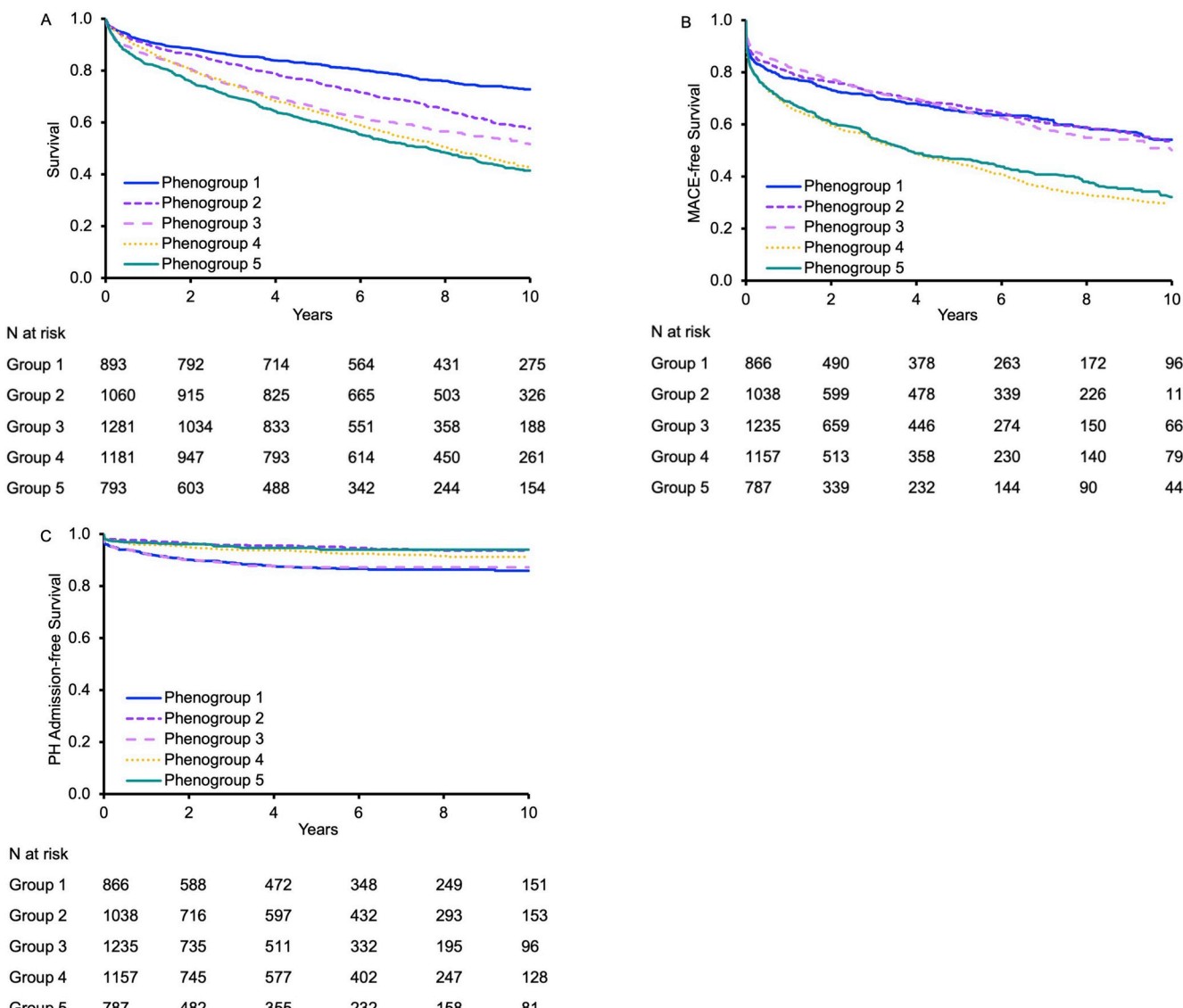

**Fig 4.** Kaplan-Meier curves for 10-year all-cause mortality (A), MACE-free survival (B) and survival free from PH admissions (C), stratified by phenogroup.

phenogroups were clinically distinct with important differences in cardiac structure and function; in contrast, hemodynamic measures were similar across phenogroups, and 3) phenogroups differed in outcomes, with phenogroup 4 ("men with cardiometabolic and CVD") having 26% higher risk of mortality and 68% higher risk of MACE compared to phenogroup 1 ("young men with obesity").

Our study in a hospital-based sample showed that patients with PH were largely older with high prevalence of multimorbidity including >50% obesity, 61% hypertension, 40% heart failure, and 31% valvular disease. The median mPAP was 29 mmHg with an elevated PCWP of 18 mmHg and a PVR of 2.2 WU. As suggested by these hemodynamic parameters, 47% of the population had isolated post-capillary PH. The low PVR and high prevalence of post-capillary disease in our sample agrees with current epidemiology showing isolated post-capillary PH due to left heart disease to be the most common [12]. In contrast, pre-capillary PH is rare and

**Table 3. Association between all-cause mortality, MACE events and PH admissions across phenogroups.**

| Phenogroup | Events/n | HR | 95% CI | P Value |
|---|---|---|---|---|
| | ALL-CAUSE MORTALITY | | | |
| 1 | 214/893 | Reference | | |
| 2 | 403/1060 | 0.7 | 0.57–0.88 | 0.002 |
| 3 | 532/1281 | 0.98 | 0.80–1.20 | 0.87 |
| 4 | 618/1181 | 1.26 | 1.04–1.52 | 0.02 |
| 5 | 415/793 | 1.14 | 0.92–1.41 | 0.25 |
| | MACE | | | |
| 1 | 293/866 | Reference | | |
| 2 | 343/1038 | 0.86 | 0.70–1.06 | 0.16 |
| 3 | 376/1235 | 0.87 | 0.72–1.06 | 0.18 |
| 4 | 609/1157 | 1.68 | 1.41–2.00 | < 0.001 |
| 5 | 381/787 | 1.52 | 1.24–1.87 | < 0.001 |
| | PH HOSPITALIZATIONS | | | |
| 1 | 98/866 | Reference | | |
| 2 | 47/1038 | 0.42 | 0.27–0.68 | < 0.001 |
| 3 | 125/1235 | 1.33 | 0.90–1.97 | 0.16 |
| 4 | 70/1157 | 0.76 | 0.51–1.12 | 0.16 |
| 5 | 36/787 | 0.67 | 0.41–1.11 | 0.12 |

CI, confidence interval; MACE, major adverse cardiac event; PH, pulmonary hypertension

represented only 14% of our population [13]. Interestingly, the hemodynamic subtype designated as other PH (PVR<3 WU and PCWP≤15 mmHg) was the second most common, occurring in 22% of the population. It is important to note that this hemodynamic phenotype is not well characterized. Recent work suggests that those with PH and PVR ≥ 2.2 are at increased risk for adverse outcomes, especially in the absence of pulmonary vascular congestion (i.e., PCWP ≤15) [14]. Based on such studies, the upper limit of normal PVR was changed in the most recent European Society of Cardiology guidelines to PVR ≤ 2 WU [15]. These findings highlight the ongoing need to better understand and refine PH hemodynamic subtypes.

Clustering analysis is an unsupervised machine learning technique that is playing an important role in understanding phenotype heterogeneity and outcome prediction across diseases [16–19]. PH is a highly heterogeneous disorder and yet to date, few studies have applied cluster-based approaches to identify high-risk subgroups. We used clustering to identify five distinct phenogroups of patients with PH. The phenogroups differed in their demographics and cardiac structure and function. In particular, phenogroup 2 ("women with hypertension") had the highest LVEF (57%) and lowest LVEDD (47 mm) as has previously been observed across sex differences in cardiac structure and function [20]. In contrast, phenogroups 4 ("men with cardiometabolic and CVD") and 5 ("men with SHD and AF") had the lowest EF (45% and 46%, respectively) and highest LVEDD (53 mm and 54 mm, respectively). With respect to PH subtypes, phenogroups 1 ("young men with obesity") and 3 ("men with overweight") had the greatest proportion of precapillary PH (17 and 22%, respectively). These findings are in keeping with previous studies demonstrating an association between obesity and cardiometabolic disease (including insulin resistance) with pulmonary vascular remodeling [21–23]. In contrast, phenogroups 4 ("men with cardiometabolic and CVD") and 5 ("men with SHD and AF") had the greatest proportion of post-capillary PH (53 and 55%, respectively), as might be expected in the setting of greater LV remodeling.

Phenogroups with greater multimorbid burden, specifically groups 4 ("men with cardiometabolic and CVD") and 5 ("men with SHD and AF"), had significantly greater risk of mortality (group 4) and MACE (both groups 4 and 5) compared with phenogroups with fewer comorbidities. Interestingly, hemodynamic parameters including mPAP and PVR were similar across both low- and high-risk groups. Thus, this work suggested that multimorbid burden determined clinical trajectory among individuals with PH. In this context, it is important to recognize that clinical trials for PH often rely on hemodynamic data as inclusion criteria, and patients with comorbidities are often excluded leading to limited data on the effectiveness of PH-specific therapies in patients with comorbid conditions [2]. While the specific etiology of PH is paramount to guiding therapies, our data also highlight the importance of multimorbidity in determining overall clinical trajectory. Whether a better understanding of multimorbid contributions, including cardiometabolic disease, to PH outcomes should prompt specific surveillance or therapies remains to be studied.

In contrast to overall and MACE-free survival, phenogroups 1 ("young men with obesity) and 3 ("men with overweight") had significantly higher risk of being admitted to the hospital with a primary diagnosis of PH in unadjusted analyses; this difference was attenuated in multivariable-adjusted models. While it is possible that overweight or obesity may lead to greater symptom burden with PH and associated admissions, this finding may also be due to competing risks, whereby older patients with multimorbidity and PH are more likely to be admitted in the setting of other causes rather than primary diagnosis of PH.

Several important limitations deserve mention. First, our study included patients undergoing clinically indicated RHC and may therefore be subject to selection bias. Second, we focused on important common clinical comorbid conditions easily abstracted from the medical record to perform clustering and acknowledge that more granular clinical or physiologic data that were not widely available may have led to more robust clustering. Specifically, we recognize that there are other clinical conditions including pulmonary thromboembolism, rheumatologic diseases, and physiologic variables including pulmonary function testing, 6-minute walk distance, and echocardiographic variables that were not included in the clustering due to low prevalence or high missingness. Third, while internal validation confirmed similar phenogroups and outcomes, we did not have access to an external patient population with undifferentiated PH for validation, limiting potential generalizability of the clusters. However, measures of cluster performance appeared reassuring including the Jaccard coefficient and PAC (though there exists no consensus on cut-off limits for stability with the latter metric). Fourth, we relied on code-based primary discharge diagnoses to define outcomes, which may have led to misclassification, though we do not believe this underestimation would result in differential bias by phenogroup. Finally, we were not able to account for PH or heart failure-specific therapies, which may have affected outcomes between clusters.

In conclusion, a cluster-based approach identified distinct multimorbid phenogroups among individuals with PH with differing clinical outcomes. Specifically, phenogroups 4 ("men with cardiometabolic and CVD") and 5 ("men with SHD and AF") had higher risk of adverse outcomes, including all-cause mortality and MACE compared to other groups. Despite higher hazard for adverse clinical outcomes, these groups did not have more clinically severe hemodynamic derangements. Overall, this study highlights the importance of multimorbidity, and in particular cardiometabolic and cardiovascular disease, in assessing prognosis in PH.

## Non-standard abbreviations and acronyms

PH: pulmonary hypertension

Cpc-PH: combined pre- and post-capillary pulmonary hypertension

CVD: cardiovascular disease

SHD: structural heart disease

PAC: proportion of ambiguous clustering

## Supporting information

**S1 Fig. Consort diagram of the study.**
(TIFF)

**S1 Table. ICD and CPT codes used to identify clinical outcomes of interest.**
(PDF)

**S2 Table. Presumed WSPH group across clusters.**
(PDF)

**S3 Table. Clinical, hemodynamic and echocardiographic characteristics stratified by phenogroup for internal validation study.**
(PDF)

**S4 Table. Association between all-cause mortality, MACE events and PH admissions across phenogroups of internal validation study.**
(PDF)

**S5 Table. Post-hoc testing for between group differences in clinical characteristics.**
(PDF)

**S6 Table. Post-hoc testing for between group differences in laboratory, hemodynamic and echocardiographic characteristics.**
(PDF)

## Author Contributions

**Conceptualization:** Paula Rambarat, Emily K. Zern, Jennifer E. Ho.

**Data curation:** Paula Rambarat, Emily K. Zern, Dongyu Wang, Carl T. Andrews, Eugene V. Pomerantsev, Jennifer E. Ho.

**Formal analysis:** Paula Rambarat, Emily K. Zern, Dongyu Wang, Athar Roshandelpoor, Shahrooz Zarbafian, Jennifer E. Ho.

**Funding acquisition:** Jennifer E. Ho.

**Investigation:** Paula Rambarat, Emily K. Zern, Jennifer E. Ho.

**Methodology:** Paula Rambarat, Emily K. Zern, Dongyu Wang, Athar Roshandelpoor, Shahrooz Zarbafian, Elizabeth E. Liu, Jessica K. Wang, Jenna N. McNeill, Nathaniel Diamant, Puneet Batra, Steven A. Lubitz, Michael H. Picard, Jennifer E. Ho.

**Project administration:** Jennifer E. Ho.

**Resources:** Jennifer E. Ho.

**Software:** Jennifer E. Ho.

**Supervision:** Michael H. Picard, Jennifer E. Ho.

**Validation:** Paula Rambarat, Emily K. Zern, Dongyu Wang, Jennifer E. Ho.

**Visualization:** Paula Rambarat, Emily K. Zern, Dongyu Wang, Elizabeth E. Liu, Jessica K. Wang, Jenna N. McNeill, Jennifer E. Ho.

**Writing – original draft:** Paula Rambarat, Emily K. Zern, Dongyu Wang, Jennifer E. Ho.

**Writing – review & editing:** Paula Rambarat, Emily K. Zern, Dongyu Wang, Athar Roshandelpoor, Elizabeth E. Liu, Jessica K. Wang, Jenna N. McNeill, Carl T. Andrews, Eugene V. Pomerantsev, Nathaniel Diamant, Puneet Batra, Steven A. Lubitz, Michael H. Picard, Jennifer E. Ho.

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
