## [Decision Letter · Decision Letter 0]

18 Jun 2023

PONE-D-23-04526

Identifying High Risk Clinical Phenogroups of Pulmonary Hypertension Through a Clustering Analysis

PLOS ONE

Dear Dr. Ho,

Thank you for submitting your manuscript to PLOS ONE. After careful consideration, we feel that it has merit but does not fully meet PLOS ONE’s publication criteria as it currently stands. Therefore, we invite you to submit a revised version of the manuscript that addresses the points raised during the review process.

We look forward to receiving your revised manuscript.

Kind regards,

Prof. Gaetano Santulli, MD, PhD

Academic Editor

PLOS ONE

“JEH was supported by NIH grants R01-HL134893, R01-HL140224, and K24-HL153669”

“JEH was supported by NIH grants R01-HL134893, R01-HL140224, and K24-HL153669.”

“JEH was supported by NIH grants R01-HL134893, R01-HL140224, and K24-HL153669”

“JEH receives research grant support from Bayer AG. PB and ND receive research support from Bayer AG and IBM research. SAL has received research support from Bristol Myers Squibb, Pfizer, Bayer AG, Boehringer Ingelheim and Fitbit, IBM, and is an employee of Novartis.”

7. We note that you have indicated that data from this study are available upon request. PLOS only allows data to be available upon request if there are legal or ethical restrictions on sharing data publicly. For more information on unacceptable data access restrictions, please see http://journals.plos.org/plosone/s/data-availability#loc-unacceptable-data-access-restrictions.

8. Thank you for stating the following in the Competing Interests section:

“JEH receives research grant support from Bayer AG. PB and ND receive research support from Bayer AG and IBM research. SAL has received research support from Bristol Myers Squibb, Pfizer, Bayer AG, Boehringer Ingelheim and Fitbit, IBM, and is an employee of Novartis.”

Reviewers' comments:

Reviewer's Responses to Questions

**Comments to the Author**

1. Is the manuscript technically sound, and do the data support the conclusions?

Reviewer #1: Partly

Reviewer #2: Yes

2. Has the statistical analysis been performed appropriately and rigorously? 

Reviewer #1: Yes

Reviewer #2: Yes

3. Have the authors made all data underlying the findings in their manuscript fully available?

Reviewer #1: Yes

Reviewer #2: Yes

4. Is the manuscript presented in an intelligible fashion and written in standard English?

Reviewer #1: Yes

Reviewer #2: Yes

5. Review Comments to the Author

Reviewer #1: I read with interest the manuscript submitted to Plos One entitled, “Identifying High Risk Clinical Phenogroups of Pulmonary Hypertension Through a Clustering Analysis.” This manuscript aims to describe phenotypes of PH from a large retrospective cohort principally through overlapping comorbidities. Then, the authors examine differential prognosis between phenotypic groups. Phenotyping is done through a directed (predefined covariates) clustering analysis and is validated with a separate internal cohort. They define 5 distinct phenotypic clusters that differ not only in comorbidities but also in hemodynamic profile and left heart structure and function. This reviewer applauds the investigators for a well-designed statistical approach. However, I have several major and minor considerations regarding this study:

Major:

1. Rationale- Although this reviewer agrees that comorbidities should have more influence in the phenotyping of patients, the investigators have chosen an approach that relies primarily on comorbidities. This reviewer is not convinced that these 5 clusters truly represent differing phenogroups. The comorbidities that were selected in combination with sex, and age would naturally predict survival. Physiologic variables (i.e., PFT, 6MWD, SpO2), imaging (i.e., CT or CXR, echocardiographic variables) and hemodynamics should be placed together with comorbidities as covariates. Was this not done due to missingness of data? If so, then this should be explained and discussed in the limitations.

2. Rationale- The investigators illustrate the clinical relevance of this analysis is based on the premise that there is “diagnostic uncertainty” in the context of WSPH G2PH when it is defined as CpC-PH (first paragraph, introduction). This reviewer agrees that there is diagnostic uncertainty when too much emphasis is placed on hemodynamic profile (i.e., iso-post cap PH, PAH, CpC-PH). In practice, this is rarely done by experienced clinicians who use a multitude of demographic, physiologic, and imaging parameters to place the hemodynamics in context.

3. Methods- Using diagnostic codes to identify clusters lacks physiologic relevance both because anecdotally codes are often mistaken and they don’t provide quantitative physiologic data, as previously mentioned. For example, a diagnosis of CKD is less meaningful than a GFR. Similarly, AHI would be of greater value than the dichotomous variable of sleep apnea. It would have made more sense to use objective data points (ie creatinine, AHI, Aortic velocity, Mitral velocity, LAVI, BMI, age, sex, HgA1C, FEV1, and liver fibrosis grade) to really delineate physiologically and clinically meaningful phenotypes. The fact that hemodynamic measures were similar across comorbid phenogroups furthers this point.

4. Methods-how is chronic heart failure, chronic lung disease defined, how was “RV dysfunction” defined in echo?

5. Methods-How is treatment especially PAH treatment or goal-directed therapy for HFrEF handled in the model? Differential therapies may affect survival between clusters.

6. Methods-should a history of PE be placed as a covariate?

7. Methods-What is T0 for time to event or censor in the cox and KM analysis?

8. Results-what are the WSPH group assignments? Do they spread across clusters? This would

9. Results-post hoc testing for Tables 1 and 2 for between group differences?

10. Discussion- The authors state that their study showed that patients with undifferentiated PH were largely older with high prevalence of multicomorbidity. I am confused by this statement as all patients should have had differentiated PH to begin with. Also, the authors do not define “undifferentiated PH”. Is this “other PH” in the PH type?

Minor:

1. This author likes the way the radar plot and violin plots (Figure 1 and 2) display the data. However, the color palate is not red-green color blind friendly. Please redo the figures without red and green on the same figure.

Reviewer #2: Overall, a very interesting paper.

Main concerns:

How were the covariates chosen? Medications, presence of PE should be included.

Some definitions are missing: How did the Authors define undifferentiated PH, chronic HF, chronic lung disease, RV dysfunction?

Please clarify WSPH group assignments.

The discussion in its present form fails to interpret the data in the context of what is known in the field: it sounds somehow redundant, as it largely summarizes again data already presented in the Results without placing them in the proper scientific context.

The paper is mainly descriptive and focused on its (not fully supported) conclusions, not adequately acknowledging the limitations of the study. The strengths and limitations of the study should be deeply addressed, taking into account sources of potential bias or imprecision: Discuss both direction and magnitude of any potential bias.

The color palette is not red-green color blind friendly. Please redo the figures without red and green on the same figure.

6. PLOS authors have the option to publish the peer review history of their article (what does this mean?). If published, this will include your full peer review and any attached files.

Reviewer #1: No

Reviewer #2: No

---

## [Author Response · Author response to Decision Letter 0]

2 Aug 2023

Reviewers’ Comments 

Reviewer 1

I read with interest the manuscript submitted to Plos One entitled, “Identifying High Risk Clinical Phenogroups of Pulmonary Hypertension Through a Clustering Analysis.” This manuscript aims to describe phenotypes of PH from a large retrospective cohort principally through overlapping comorbidities. Then, the authors examine differential prognosis between phenotypic groups. Phenotyping is done through a directed (predefined covariates) clustering analysis and is validated with a separate internal cohort. They define 5 distinct phenotypic clusters that differ not only in comorbidities but also in hemodynamic profile and left heart structure and function. This reviewer applauds the investigators for a well-designed statistical approach. However, I have several major and minor considerations regarding this study:

Major:

1. Rationale- Although this reviewer agrees that comorbidities should have more influence in the phenotyping of patients, the investigators have chosen an approach that relies primarily on comorbidities. This reviewer is not convinced that these 5 clusters truly represent differing phenogroups. The comorbidities that were selected in combination with sex, and age would naturally predict survival. Physiologic variables (i.e., PFT, 6MWD, SpO2), imaging (i.e., CT or CXR, echocardiographic variables) and hemodynamics should be placed together with comorbidities as covariates. Was this not done due to missingness of data? If so, then this should be explained and discussed in the limitations. 

RESPONSE: We thank the Reviewer for this thoughtful comment. We focused on comorbidities for several reasons: first, clustering on clinical comorbidies alone may be more accessible to clinicians as this information is readily available in the clinic setting when initially meeting a patient with PH, which has not yet been further characterized. Second, we acknowledge that other physiologic data outlined by the Reviewer (PFT, 6MWD, imaging) was not widely available, and that high degree of missingness and ascertainment bias would have reduced the sample size significantly and precluded cluster-based analyses. We have added this point in the limitations section as suggested by the Reviewer. 

Page 13, paragraph 3: “Second, we focused on important common clinical comorbid conditions easily abstracted from the medical record to perform clustering and acknowledge that more granular clinical or physiologic data that were not widely available may have led to more robust clustering. Specifically, we recognize that there are other clinical conditions including pulmonary thromboembolism, rheumatologic disease, and physiologic variables including pulmonary function testing, 6-minute walk distance, and echocardiographc variables that were not included in the clustering due to low prevalence or high missingness.” 

2. Rationale- The investigators illustrate the clinical relevance of this analysis is based on the premise that there is “diagnostic uncertainty” in the context of WSPH G2PH when it is defined as CpC-PH (first paragraph, introduction). This reviewer agrees that there is diagnostic uncertainty when too much emphasis is placed on hemodynamic profile (i.e., iso-post cap PH, PAH, CpC-PH). In practice, this is rarely done by experienced clinicians who use a multitude of demographic, physiologic, and imaging parameters to place the hemodynamics in context. 

RESPONSE: We thank the Reviewer for this comment. The authors agree that classification of WSPH groups is complex, and that in practice, experienced clinicians are integrating many complementary pieces of information, of which hemodynamic data is only one piece. Our analysis illustrates that in addition to hemodynamic heterogeneity, there is much overlap in comorbid diagnoses which make it more challenging to contextualize hemodynamics. We have amended the introduction to emphasize this important point made by the Reviewer. 

Page 4, paragraph 1: “In practice, however, multimorbidity including concomitant cardiopulmonary disease is common which can make defining the etiology of elevated pulmonary pressures more challenging. In addition, overlap between WSPH groups often exists.”

3. Methods- Using diagnostic codes to identify clusters lacks physiologic relevance both because anecdotally codes are often mistaken and they don’t provide quantitative physiologic data, as previously mentioned. For example, a diagnosis of CKD is less meaningful than a GFR. Similarly, AHI would be of greater value than the dichotomous variable of sleep apnea. It would have made more sense to use objective data points (ie creatinine, AHI, Aortic velocity, Mitral velocity, LAVI, BMI, age, sex, HgA1C, FEV1, and liver fibrosis grade) to really delineate physiologically and clinically meaningful phenotypes. The fact that hemodynamic measures were similar across comorbid phenogroups furthers this point. 

RESPONSE: We thank the Reviewer for this salient point. The authors agree that using quantitative objective data for clustering or defining phenotypes may have improved ascertainment of comorbid conditions beyond code-based diagnoses. We would like to clarify that major comorbidities including hypertension, heart failure, chronic lung disease, renal failure history, diabetes, and prior coronary artery disease were clinically adjudicated, rather than based on billing or procedure codes. By contrast, obstructive sleep apnea and atrial fibrillation were obtained using billing codes. We agree with the Reviewer that clinically adjudicated comorbidities are likely more accurate than code-based diagnoses, and now clarify ascertainment in the methods section. Further, we examined other physiologic data outlined by the Reviewer -we have summarized missingness in Reviewer Table 1 below, and acknowledge that clustering is limited as these granular data are not widely available. We have amended the limitations section to include this important point.

Page 5, paragraph 1: “Clinical characteristics, laboratory data, and comorbidities were ascertained from the medical history as part of clinical assessment done at the time of RHC, except for obstructive sleep apnea (OSA) and atrial fibrillation (AF), which were identified based on International Classification of Diseases Ninth Revision (ICD‐9) or Tenth Revision (ICD‐10) codes.”

Page 13, paragraph 3: “… we focused on important common clinical comorbid conditions easily abstracted from the medical record to perform clustering and acknowledge that more granular clinical or physiologic data that were not widely available may have led to more robust clustering.”

Reviewer Table 1: Missingness in physiologic quantitative parameters 

Variables Missingness (%) N*

Hemoglobin 85 782

Hemoglobin A1C 45 2882

Total cholesterol 33 3502

LDL 35 3380

HDL 33 3488

NT-pro BNP 46 2821

Mean gradient across MV 92 427

Mean gradient across AV 81 1007

*N is number of samples with extracted data out of 5208 total participants. 

4. Methods-how is chronic heart failure, chronic lung disease defined, how was “RV dysfunction” defined in echo? 

RESPONSE: Thank you – we are happy to clarify. Chronic heart failure and chronic lung disease were ascertained from the medical record as part of routine data collected at the time of the RHC. RV dysfunction was defined based on clinical read of RV dysfunction of at least moderate severity by a sonographer or cardiologist. For clarity, we have included these definitions in the methods section, under subtitle data collection.

Page 5, paragraph 1: “Clinical characteristics, laboratory data, and comorbidities were ascertained from the medical history as part of clinical assessment done at the time of RHC… RV dysfunction was defined based on clinical report of RV dysfunction of at least moderate severity.”

5. Methods-How is treatment especially PAH treatment or goal-directed therapy for HFrEF handled in the model? Differential therapies may affect survival between clusters. 

RESPONSE: We thank the Reviewer for this astute comment. Unfortunately, PAH and HF treatment was not routinely ascertained at time of the RHC and thus we are not able to include an analysis of these therapies in our study. We have noted in our limitations section the possibility that differential therapies may have affected survival between clusters.

Page 13, paragraph 3: “Finally, we were not able to account for PH or heart failure-specific therapies, which may have affected outcomes between clusters.”

6. Methods-should a history of PE be placed as a covariate? 

RESPONSE: We thank the Reviewer for this suggestion. History of PE is certainly an important covariate to consider in patients with PH, especially given its association with development of CTEPH. We performed further analyses to abstract prevalent PE from the medical record using appropriate ICD-9 or ICD-10 codes. We found 150 patients in our sample had a history of PE. In an exploratory analysis when added to our clustering model, all patients with PE were found to cluster in one group, cluster 3 (Reviewer Table 2). When comparing these new clusters to the original clusters, we found that cluster 3 was intermediate between previous cluster 1 and cluster 3 and notable for male-predominant individuals with higher BMI and mix of cardiometabolic comorbidities. Given the low prevalence of PE in the overall population, we favor not including this covariate as a clustering input because it shifted all individuals with PE into one cluster irrespective of other comorbid conditions and unduly influences clustering. We have included prior PE as a clinical covariate in Table 1 of the manuscript as requested by the Reviewer. 

Reviewer Table 2: Clinical characteristics (including PE) stratified by phenogroup 

Phenogroup Total

(n=5208) 1

(n=1597) 2

(n=1682) 3

(n=150) 4

(n=719) 5

(n=1060) p value

Demographics

Age, years 64 (12) 55 (13) 70 (7) 62 (13) 66 (10) 67 (9) < 0.001

Male sex 3176 (61) 1165 (73) 402 (24) 83 (55) 642 (89) 884 (83) < 0.001

BMI (kg/m2) 30.2 (7.3) 29.0 (7.0) 29.6 (7.2) 31.3 (8.5) 33.3 (8.1) 30.7 (6.5) < 0.001

Clinical Comorbidities

Hypertension 3150 (61) 250 (16) 1241 (74) 60 (40) 620 (86) 979 (92) < 0.001

Diabetes Mellitus 1401 (27) 119 (8) 318 (19) 29 (19) 159 (22) 776 (73) < 0.001

Prior MI 1079 (21) 77 (5) 163 (10) 23 (15) 60 (8) 756 (71) < 0.001

Heart failure 2057 (40) 291 (18) 355 (21) 43 (29) 509 (71) 859 (81) < 0.001

Valvular disease 1884 (36) 409 (26) 701 (42) 34 (23) 313 (44) 427 (40) < 0.001

AF 1128 (22) 256 (16) 372 (22) 38 (25) 230 (32) 232 (22) < 0.001

OSA 868 (17) 171 (11) 134 (8) 33 (22) 461 (64) 69 (7) < 0.001

Chronic lung disease 941 (18) 164 (10) 331 (20) 31 (21) 165 (23) 

250 (24) < 0.001

CKD 201 (4) 17 (1) 42 (3) 9 (6) 38 (5) 95 (9) < 0.001

PE 150 (3) 0 0 150 (100) 0 0 < 0.001

7. Methods-What is T0 for time to event or censor in the cox and KM analysis? 

RESPONSE: Thank you for the opportunity to clarify. T0 for time to event or censoring in the Cox and KM analysis was the date of right heart catheterization. For clarity, this has been added to the methods section. 

Page 6, paragraph 2: “For time-to-event analyses, start of follow-up was the date of RHC.”

8. Results-what are the WSPH group assignments? Do they spread across clusters? 

RESPONSE: We appreciate the opportunity to expand upon this point. Specific WSPH group assignments were unfortunately not widely available across such a large hospital-based sample. We agree that it would be interesting to characterize how WSPH groups spread across clusters and have pursued additional analyses to examine comorbid conditions that may provide further information as to probable WSPH group. In Reviewer Table 3, we summarize probable WSPH groups for all individuals with PH based on clusters using the following definitions: group 1: connective tissue disease, group 2: left heart disease (defined as heart failure, previous MI or previous AF), group 3: chronic lung disease or OSA, and group 4: PE. We also display overlap and recognize interestingly that more than half of individuals in clusters 1, 2, and 3 have overlapping WSPH group criteria, whereas clusters 4 and 5 are predominantly marked by left heart disease. We recognize that this analysis is limited as it relies solely on comorbid data ascertained from medical records to define WSPH group but may offer some insight into how WSPH groups varied across clusters. Although limited, this analysis illustrates multimorbidity in our population which can make it difficult to place a patient with PH into one exclusive WSPH group. We have included this information as Supplemental Table 2 in the manuscript. 

Reviewer Table 3: Presumed WSPH group across clusters 

Presumed WSPH Group Cluster 1 (%) Cluster 2 (%) Cluster 3 (%) Cluster 4 (%) Cluster 5 (%)

Group 1 2.2 0.9 1.3 0.0 0.0

Group 2 27.3 36.5 23.1 64.4 81.5

Group 3 9.7 6.8 9.9 5.2 0.0

Group 4 2.2 0.7 2.3 0.3 0.0

Groups 1&2 0.7 0.6 0.6 0.6 0.8

Groups 1&3 0.6 0.3 0.3 0.0 0.0

Groups 1&4 0.1 0.0 0.2 0.0 0.0

Groups 2&3 4.9 3.7 6.9 15.3 14.6

Groups 2& 4 0.1 0.0 0.1 0.1 0.1

3 or more groups 52.1 50.7 55.4 14.1 3.0

Page 9, paragraph 1: “When examining presumed WSPH group (based on presence of connective tissue disease (group 1), left heart disease (group 2), lung disease (group 3), and PE (group 4), we found that over 50% of patients in clusters 1, 2 , and 3 met criteria for 3 or more WSPH groups, whereas the majority of patients in clusters 4 and 5 had left heart disease (Supplemental Table 2).” 

9. Results-post hoc testing for Tables 1 and 2 for between group differences?

RESPONSE: Please find below post-hoc testing for Manuscript Table 1 and Manuscript Table 2 for between group differences using cluster 1 as reference. We have included these in Supplemental Tables 5 and 6 in the manuscript. 

Reviewer Table 4: Post-hoc testing for between group differences in clinical characteristics 

Variable Reference Comparison p-value

 Cluster 2 Cluster 3 Cluster 4 Cluster 5

Demographics 

Age Cluster 1 <0.0001 <0.0001 <0.0001 <0.0001

Male sex Cluster 1 <0.0001 0.8592 <0.0001 <0.0001

BMI Cluster 1 0.0466 0.1384 <0.0001 <0.0001

Clinical Comorbidities 

Hypertension Cluster 1 <0.0001 0.0393 <0.0001 <0.0001

Diabetes Mellitus Cluster 1 <0.0001 <0.0001 <0.0001 <0.0001

Prior MI Cluster 1 <0.0001 0.1141 <0.0001 <0.0001

Heart failure Cluster 1 0.3050 <0.0001 <0.0001 <0.0001

Valvular disease Cluster 1 <0.0001 <0.0001 0.8525 <0.0001

AF Cluster 1 0.4955 <0.0001 0.1012 <0.0001

OSA Cluster 1 0.0011 0.1952 0.0021 0.5274

Chronic lung disease Cluster 1 <0.0001 0.0172 <0.0001 <0.0001

CKD Cluster 1 0.2165 0.3366 <0.0001 0.0001

Reviewer Table 5: Post-hoc testing for between group differences in laboratory, hemodynamic and echocardiographic characteristics 

Variable Reference Comparison p-value

 Cluster 2 Cluster 3 Cluster 4 Cluster 5

Laboratory 

NT-pro BNP (pg/ml) Cluster 1 0.0554 0.5476 0.0173 <0.0001

Hemodynamics 

Mean PAP (mmHg) Cluster 1 <0.0001 0.9275 0.9961 0.2230

Mean RAP (mmHg) Cluster 1 <0.0001 0.0850 0.0327 0.4546

PCWP (mmHg) Cluster 1 0.3528 0.014 <0.0001 <0.0001

TPG (mmHg) Cluster 1 0.0022 0.0007 0.0577 0.0004

PVR (WU) Cluster 1 0.9584 <0.0001 0.7467 0.7973

PAPi Cluster 1 <0.0001 <0.0001 0.7012 0.0402

TD CI (L/min/m2) Cluster 1 <.00001 <0.0001 <0.0001 <0.0001

Pre-capillary PH * Cluster 1 <0.0001 0.0008 0.0006 <0.0001

Post-capillary PH * Cluster 1 

Cpc-PH * Cluster 1 

Other PH * Cluster 1 

Echocardiography 

LVEF (%) Cluster 1 <0.0001 <0.0001 <0.0001 0.0069

LVEDD (mm) Cluster 1 <0.0001 0.0001 <0.0001 <0.0001

RV systolic dysfunction (%) Cluster 1 0.5495 0.0001 0.8166 0.4516

MR (%) Cluster 1 <0.0001 0.7625 0.7438 <0.0001

TR (%) Cluster 1 0.9188 0.0006 0.6059 <0.0001

*Analysis performed using cumulative logit model 

10. Discussion- The authors state that their study showed that patients with undifferentiated PH were largely older with high prevalence of multimorbidity. I am confused by this statement as all patients should have had differentiated PH to begin with. Also, the authors do not define “undifferentiated PH”. Is this “other PH” in the PH type? 

RESPONSE: We thank the Reviewer for this comment. Our analysis included all patients who underwent a RHC at our institution and were then found to have PH as defined by a mean pulmonary artery pressure > 20 mmHg. As such, the analysis was not limited to patients with an established diagnosis of PH. The authors agree that the use of the word undifferentiated in the first sentence of the second paragraph of our discussion is unclear and we have therefore removed it from the sentence. 

Minor:

1. This author likes the way the radar plot and violin plots (Figure 1 and 2) display the data. However, the color palate is not red-green color blind friendly. Please redo the figures without red and green on the same figure. 

RESPONSE: We thank the Reviewer for this important comment. All figures have been redone to be red-green color blind friendly.

 

Reviewer 2

Overall, a very interesting paper.

How were the covariates chosen? Medications, presence of PE should be included.

RESPONSE: We thank the Reviewer for this suggestion. Covariates were chosen based on major clinical comorbid conditions, and were chosen for clustering as these variables are easily accessible to clinicians who may be seeing a patient in clinic whose PH has not yet been fully characterized or risk stratified. We agree that PE is an important covariate to consider in patients with PH, especially given its association with development of CTEPH. We abstracted additional data on prevalent PE using appropriate ICD-9 or ICD-10 codes. Only 150 patients in our sample had a prevalent history of PE at time of the RHC, we have included this in Table 1 of the manuscript. Unfortunately, PAH and HF medications were not routinely ascertained at time of the RHC, which we have noted in the limitations.

Page 13, paragraph 3: “we were not able to account for PH or heart failure-specific therapies, which may have affected outcomes between clusters.”

Some definitions are missing: How did the Authors define undifferentiated PH, chronic HF, chronic lung disease, RV dysfunction?

RESPONSE: We thank the Reviewer for this comment. Undifferentiated PH (as used in the second paragraph of the discussion) is patients with PH who have yet to be characterized into a hemodynamic profile or WSPH group. We note that the use of the word undifferentiated is unclear and we have therefore removed it from the sentence. History of chronic heart failure and chronic lung disease was ascertained by a clinician as part of routine data collected at the time of the RHC. RV dysfunction was defined based on a clinical read of RV dysfunction of at least moderate severity. For clarity, we will includes these definitions in the methods section, under subtitle data collection. 

Page 5, paragraph 1: “Clinical characteristics, laboratory data, and comorbidities were ascertained from the medical history as part of clinical assessment done at the time of RHC… RV dysfunction was defined based on clinical report of RV dysfunction of at least moderate severity.”

Please clarify WSPH group assignments.

RESPONSE: We appreciate the opportunity to expand upon this point. Specific WSPH group assigments were unfortunately not widely available across such a large hospital-based sample. We agree that it would be interesting to characterize how WSPH groups spread across clusters and have pursued additional analyses to examine comorbid conditions that may provide further information as to probable WSPH group. In Reviewer Table 3, we summarize probable WSPH groups for all individuals with PH based on clusters using the following definitions: group 1: connective tissue disease, group 2: left heart disease (defined as heart failure, previous MI or previous AF), group 3: chronic lung disease or OSA, and group 4: PE. We also display overlap and recognize interestingly that more than half of individuals in clusters 1, 2, and 3 have overlapping WSPH group criteria, whereas clusters 4 and 5 are predominantly marked by left heart disease. We recognize that this analysis is limited as it relies solely on comorbid data ascertained from medical records to define WSPH group but may offer some insight into how WSPH groups varied across clusters. Although limited, this analysis illustrates multimorbidity in our population which can make it difficult to place a patient with PH into one exclusive WSPH group. We have included this information as Supplemental Table 2 in the manuscript. 

Reviewer Table 3: Presumed WSPH group across clusters 

Presumed WSPH Group Cluster 1 (%) Cluster 2 (%) Cluster 3 (%) Cluster 4 (%) Cluster 5 (%)

Group 1 2.2 0.9 1.3 0.0 0.0

Group 2 27.3 36.5 23.1 64.4 81.5

Group 3 9.7 6.8 9.9 5.2 0.0

Group 4 2.2 0.7 2.3 0.3 0.0

Groups 1&2 0.7 0.6 0.6 0.6 0.8

Groups 1&3 0.7 0.3 0.3 0.0 0.0

Groups 1&4 0.1 0.0 0.2 0.0 0.0

Groups 2&3 4.9 3.7 6.9 15.3 14.6

Groups 2& 4 0.1 0.0 0.1 0.1 0.1

3 or more groups 52.1 50.7 55.4 14.1 3.0

Page 9, paragraph 1: “When examining presumed WSPH group (based on presence of connective tissue disease (group 1), left heart disease (group 2), lung disease (group 3), and PE (group 4), we found that over 50% of patients in clusters 1, 2 , and 3 met criteria for 3 or more WSPH groups, whereas the majority of patients in clusters 4 and 5 had left heart disease (Supplemental Table 2.)” 

The discussion in its present form fails to interpret the data in the context of what is known in the field: it sounds somehow redundant, as it largely summarizes again data already presented in the Results without placing them in the proper scientific context. The paper is mainly descriptive and focused on its (not fully supported) conclusions, not adequately acknowledging the limitations of the study. The strengths and limitations of the study should be deeply addressed, taking into account sources of potential bias or imprecision: Discuss both direction and magnitude of any potential bias. 

RESPONSE: We thank the Reviewer for this comment. The authors acknowledge that our study has several limitations that must be taken into account when interpreting the results. For example, major comorbidities were clinically adjudicated, though more precise information including labs (eGFR, A1c, medications) were not taken into account and may have led to misclassification of comorbid conditions. We also were limited to widely available data across a large hospital-based sample to conduct cluster-based analyses, and acknowledge that more granular phenotyping including for example 6-minute walk distance, lung function testing, and other physiologic information may have helped refine clusters. We have expanded the limitations section of our discussion to include additional sources of potential bias and imprecision as outlined above. 

Page 13, paragraph 3: “… we focused on important common clinical comorbid conditions easily abstracted from the medical record to perform clustering and acknowledge that more granular clinical or physiologic data that were not widely available may have led to more robust clustering. Specifically, we recognize that there are other clinical conditions including pulmonary thromboembolism, rheumatologic disease, and physiologic variables including pulmonary function testing, 6-minute walk distance, and echocardiographic variables that were not included in the clustering due to low prevalence or high missingness.”

The color palette is not red-green color blind friendly. Please redo the figures without red and green on the same figure. 

RESPONSE: We thank the Reviewer for this important comment. All figures have been redone to be red-green color blind friendly.

---

## [Editor Report · Decision Letter 1]

10 Aug 2023

Identifying High Risk Clinical Phenogroups of Pulmonary Hypertension Through a Clustering Analysis

PONE-D-23-04526R1

Dear Dr. Ho,

We’re pleased to inform you that your manuscript has been judged scientifically suitable for publication and will be formally accepted for publication once it meets all outstanding technical requirements.

Kind regards,

Gaetano Santulli, MD

Academic Editor

PLOS ONE

---

## [Editor Report · Acceptance letter]

17 Aug 2023

PONE-D-23-04526R1 

Identifying high risk clinical phenogroups of pulmonary hypertension through a clustering analysis 

Dear Dr. Ho:

I'm pleased to inform you that your manuscript has been deemed suitable for publication in PLOS ONE. Congratulations! Your manuscript is now with our production department. 

Kind regards, 

on behalf of

Professor Gaetano Santulli 

Academic Editor

PLOS ONE